REGISTERED REPORT

# Registered report: Diverse somatic mutation patterns and pathway alterations in human cancers

Vidhu Sharma[1], Lisa Young[1], Anne B Allison[2], Kate Owen[3], Reproducibility Project: Cancer Biology*

[1]Applied Biological Materials, Richmond, Canada; [2]Piedmond Virginia Community College, Charlottesville, United States; [3]University of Virginia, Charlottesville, United States

**Abstract** The Reproducibility Project: Cancer Biology seeks to address growing concerns about reproducibility in scientific research by conducting replications of selected experiments from a number of high-profile papers in the field of cancer biology. The papers, which were published between 2010 and 2012, were selected on the basis of citations and Altmetric scores (*Errington et al., 2014*). This Registered Report describes the proposed replication plan of key experiments from "Diverse somatic mutation patterns and pathway alterations in human cancers" by Kan and colleagues published in *Nature* in 2010 (*Kan et al., 2010*). The experiments to be replicated are those reported in Figures 3D-F and 4C-F. Kan and colleagues utilized mismatch repair detection (MRD) technology to identify somatic mutations in primary human tumor samples and identified a previously uncharacterized arginine 243 to histidine (R243H) mutation in the G-protein α subunit *GNAO1* in breast carcinoma tissue. In Figures 3D-F, Kan and colleagues demonstrated that stable expression of mutant *GNAO1$^{R243D}$* conferred a significant growth advantage in human mammary epithelial cells, confirming the oncogenic potential of this mutation. Similarly, expression of variants with somatic mutations in *MAP2K4*, a JNK pathway kinase (shown in Figures 4C-E) resulted in a significant increase in anchorage-independent growth. Interestingly, these mutants exhibited reduced kinase activity compared to wild type *MAP2K4*, indicating these mutations impose a dominant-negative influence to promote growth (Figure 4F). The Reproducibility Project: Cancer Biology is a collaboration between the Center for Open Science and Science Exchange and the results of the replications will be published in *eLife*.

*For correspondence: nicole@scienceexchange.com

Group author details: Reproducibility Project: Cancer Biology See page 20

## Introduction

Human cancer is driven by the acquisition of mutations in cells of somatic origin. Somatic mutations comprise several distinct classes of DNA sequence changes, including single-nucleotide substitutions, small insertions and deletions (indels), copy number alterations, and structural rearrangements (*Weir et al., 2007*; *Chin and Gray, 2008*; *Stratton et al., 2009*; *Pleasance et al., 2010*). Somatic mutations can be further characterized based on their oncogenic ability: genetic variations that are directly involved in cancer development are termed "driver" mutations, whereas mutations that do not confer any obvious advantage are referred to as "passenger" mutations (*Davies et al., 2005*). In all cases, genetic changes in somatic cells arise as a result of defective DNA repair mechanisms and/or imprecise DNA replication, and can develop spontaneously, be acquired over the lifetime of an individual, or by direct exposure to mutagens, such as tobacco smoke and ionizing UV radiation (*Pfeifer, 2010*; *Pleasance et al., 2010*; *Helleday et al., 2014*). Over the past 10 years, technologies for the detection of wide-spread genetic alterations have been developed and used to analyze

cancer genomes (*Stratton et al., 2009*; *Watson et al., 2013*). Its is clear that cancer cell genomes often harbor substantial somatic mutation burdens, thus the ability to generate a comprehensive genetic cancer profile has the potential to significantly improve patient diagnosis and treatment.

The combination of PCR and Sanger sequencing to identify mutations in tumor genomes has proven to be a powerful approach in the study of cancer genomics (*Collins et al., 2003*). However, this technology is constrained by limited throughput and cost (*Chin et al., 2011*). Here, Kan and colleagues utilized mismatch repair detection (MRD) technology as a low-cost, high throughput alternative to identify somatic mutations in a large number of primary human tumor samples (*Peters et al., 2007*). Using this technique, Kan and colleagues identified an uncharacterized somatic mutation in *GNAO1* from breast carcinoma tissue (*Kan et al., 2010*). *GNAO1* encodes the Gαo subunit of heterotrimeric guanine-binding proteins (G proteins) (*Jastrzebska, 2013*). G proteins function as molecular switches that alternate between "on" (GTP-bound) and "off" (GDP-bound) states to control signal transduction in eukaryotes (*Gilman, 1987*; *Birnbaumer, 2007b*; *2007a*). While previous studies have reported oncogenic mutations in the Gα subunits of other G proteins, including *GNAS*, *GNAI2* and *GNAQ* (*Landis et al., 1989*; *Lyons et al., 1990*; *Forbes et al., 2008*; *Van Raamsdonk et al., 2009*), the arginine 243 to histidine (R243H) conversion identified in *GNAO1* does not correspond to any previously described mutations within G proteins (*Garcia-Marcos et al., 2011*). In Figure 3D–F, the oncogenic potential of this mutation was tested. Human mammary epithelial cells (HMECs) stably expressing equivalent levels of wild type *GNAO1* or *GNAO1^R243H* were suspended in agar before assessment for colony formation. This key experiment reported that the R243H mutation promotes a two-fold increase in anchorage-independent growth compared to cells expressing wild type *GNAO1*, and will be replicated in Protocol 1. Subsequent work on *GNAO1* has characterized the molecular basis underlying the oncogenic properties of the R243H mutation. Importantly, these studies have determined that the R243H mutation renders Gαo constitutively active via Src-STAT3 signaling (*Garcia-Marcos et al., 2011*; *Leyme et al., 2014*).

Kan and colleagues also identified a number of somatic mutations in mitogen activated protein kinase kinase 4 (*MAP2K4*) (*Kan et al., 2010*). MAP2K4 is a component of a triple kinase cascade that involves the successive activation of downstream MAP kinases, culminating in the activation of c-Jun NH2-terminal kinases (JNK) and p38 (*Derijard et al., 1995*; *Chang and Karin, 2001*; *Johnson and Lapadat, 2002*). Both the JNK and p38 signaling pathways mediate cellular responses to cytokine signals, stress and other extracellular stimuli (*Johnson and Lapadat, 2002*). While mutations in *MAP2K4* have been reported here (*Kan et al., 2010*) and elsewhere (*Teng et al., 1997*; *Parsons et al., 2005*; *Greenman et al., 2007*; *Forbes et al., 2008*), the role of *MAP2K4* in cancer has remained complex and contradictory. Some studies have suggested *MAP2K4* functions as a pro-oncogenic molecule in breast and pancreatic tumors (*Wang et al., 2004*), melanoma (*Finegan and Tournier, 2010*), and in prostate cancer tumors (*Lotan et al., 2007*; *Pavese et al., 2014*), whereas other early reports identified *MAP2K4* as a putative tumor suppressor gene due to its frequent inactivation in human cancer cell lines and tumor tissues, including pancreatic, breast, ovarian, and colon cancer cells and tissues (*Su et al., 1998*; *2002*; *Nakayama et al., 2006*; *Ahn et al., 2011*).

In Figure 4C–E, the functional relevance of six select *MAP2K4* mutants (5 located in the kinase domain, 1 outside the kinase domain) were tested *in vitro* (*Kan et al., 2010*). NIH3T3 fibroblasts stably expressing equivalent levels of either WT or mutant *MAP2K4* were assessed for their ability to promote anchorage-independent growth. Importantly, all six *MAP2K4* variants resulted in significantly enhanced agar colony formation compared to cells expressing wild type *MAP2K4*. A majority of the *MAP2K4* mutants resulted in reduced activity to either JNK or myelin basic protein (MBP) when tested in an *in vitro* kinase assay suggesting that reduced MAP2K4 signaling plays a dominant-negative role in the control of cell growth. A related study examined the invasiveness of cells where endogenous *MAP2K4* was depleted and various *MAP2K4* mutants were added back, including four of the mutants tested by Kan and colleagues (*Ahn et al., 2011*). The effect on invasion was directly proportional to the kinase activities of the mutants. The mutations that resulted in loss-of-function kinase activity (including R154W, S251N, and N234I examined by Kan and colleagues) resulted in increased invasion, while mutations with gain-of-function kinase activity, or comparable kinase activity to wild-type (including A279T examined by Kan and colleagues), did not (*Ahn et al., 2011*). More recent studies have confirmed these findings, showing that *MAP2K4* genetic inactivation is prevalent in high grade serous and endometrioid carcinomas, breast cancer, and pancreatic cancer (*Davis et al., 2011*; *Yeasmin et al., 2011b*; *Yeasmin et al., 2011a*; *Curtis et al., 2012*;

*Huang et al., 2013*). Furthermore, genetic polymorphisms that increase *MAP2K4* promoter activity are associated with reduced risk of prostate, lung, and sporadic colorectal cancers (*Wei et al., 2009*; *Liu et al., 2010*; *Shao et al., 2012*). A recent study by Haeusgen and colleagues (*Haeusgen et al., 2014*) suggests that the balance between *MAP2K4* and a novel *MAP2K4* splice variant may be important in regulating appropriate cell growth. The key experiments described in Figures 4C–F will be replicated in Protocol 2.

# Materials and methods

Unless otherwise noted, all protocol information was derived from the original paper, references from the original paper, or information obtained directly from the authors. An asterisk (*) indicates data or information provided by the Reproducibility Project: Cancer Biology core team. A hashtag (#) indicates information provided by the replicating lab.

### Protocol 1: Generation of N-terminally Flag-tagged *MAP2K4* and *GNAO1* wild-type and mutant vectors

This protocol generates N-terminally flag-tagged wild type or mutant *GNAO1* and wild type or mutant *MAP2K4* vectors. These vectors will be used in Protocols 2 and 4.

## Sampling

- This experiment will be performed once in order to generate vectors.

## Materials and reagents

| Reagent | Type | Manufacturer | Catalog # | Comments |
|---|---|---|---|---|
| pRetroX-IRES-ZsGreen1 Vector | Plasmid | Clontech | 632520 | Original product number not specified; replaces pRetro-IRES-GFP-Vector |
| *MAP2K4*$^{WT}$ Myc-DDK tagged –includes FLAG tag[1] | Plasmid | Origene | RC206051 | Original product number not specified |
| *GNAO1*$^{WT}$ Myc-DDK tagged (Variant 1) – includes FLAG tag[1] | Plasmid | Origene | RC217958 | Original product number not specified |
| Agilent - QuikChange Lightning Multi Site-Directed Mutagenesis Kit | Kit | Agilent | 210516 | Original product number not specified |

[1]DDK is equivalent to FLAG which is a registered trademark of Sigma Aldrich

## Procedure

1. 1. Generate *GNAO1* and *MAP2K4* mutant constructs:
    a. Perform site-directed mutations on cDNA ORFs using #Agilent QuikChange Kit according to manufacturer's protocol.
        i. Point mutations:
            1. *GNAO1*: arginine 243 to histidine (R243H)
            2. *MAP2K4*: arginine 228 to lysine (R228K)
            3. *MAP2K4*: alanine 279 to threonine (A279T)
2. Clone inserts (includes FLAG tag) into #pRetroX-IRES-ZsGreen1 vector backbone according to manufacturer's protocols.
    a. Specific molecular cloning steps and reagents used will be recorded and reported later.
    b. #Perform PCR cloning using primers that encompass the ORF and FLAG-tag insert from the original cDNA
3. Sequence vectors to confirm identity as well as mutational status, and run on gel to confirm integrity. [additional QC]
    a. #Use the following sequencing primers:
        i. *GNAO1*$^{R243H}$ Forward: GCCCTTTTTGAGTTTGGATC
        ii. *GNAO*$^{R243H}$ Reverse: GTAAAGCATGTGCACCGAGG
        iii. *MAP2K4*$^{R228K}$ Forward: GCCCTTTTTGAGTTTGGATC
        iv. *MAP2K4*$^{R228K}$ Reverse: GTAAAGCATGTGCACCGAGG
        v. *MAP2K4*$^{A279T}$ Forward: GCCCTTTTTGAGTTTGGATC
        vi. *MAP2K4*$^{A279T}$ Reverse: GTAAAGCATGTGCACCGAGG

## Deliverables

- Data to be collected
  - Sequencing information and gel verification of vectors

- Sample delivered for further analysis:
  - Plasmids for use in Protocols 2 and 4:
    - pRetroX-IRES-ZsGreen1
    - pRetroX-FLAG-$GNAO1^{WT}$-IRES-ZsGreen1
    - pRetroX-FLAG-$GNAO1^{R243H}$-IRES-ZsGreen1
    - pRetroX-FLAG-$MAP2K4^{WT}$-IRES-ZsGreen1
    - pRetroX-FLAG-$MAP2K4^{R228K}$-IRES-ZsGreen1
    - pRetroX-FLAG-$MAP2K4^{A279T}$-IRES-ZsGreen1

## Confirmatory analysis plan

- None applicable.

## Known differences from the original study

The vector backbone pRetroX-IRES-ZsGreen1 will be used instead of pRetroX-IRES-FLAG because the latter is no longer available. The replicating lab will use a cDNA with an ORF tagged with myc-DDK (the same as FLAG) for downstream protocols. Not all mutants used in the original study will be replicated. We will not generate MAP2K4 mutations G85R, R154W, N234I or S251N. All known differences are listed in the materials and reagents section above with the originally used item listed in the comments section. All differences have the same capabilities as the original and are not expected to alter the experimental design.

## Provisions for quality control

Sequencing and gel analysis of plasmids will be reported. All of the raw data, including the analysis files, will be uploaded to the project page on the OSF (https://osf.io/jpeqg/) and made publically available.

## Protocol 2: Generation of human mammary epithelial cells stably expressing wild-type or GNAO1$^{R243H}$

This protocol describes the generation of HMECs stably expressing WT or mutant GNAO1$^{R243H}$ protein. Expression of GNAO1 will be confirmed by Western blot that will be a replication of Figure 3F. These cells will subsequently be used in Protocol 3.

## Sampling

- This experiment to be conducted one time to confirm stable expression of GNAO1$^{WT}$ or GNAO1$^{R243H}$ protein.
- The experiment has 4 cohorts:
  - Cohort 1: Uninfected HMECs [additional negative control]
  - Cohort 2: HMECs transduced with pRetroX-IRES-ZsGreen1 -empty vector [additional negative control]
  - Cohort 3: HMECs transduced with pRetroX-FLAG-$GNAO1^{WT}$-IRES-ZsGreen1
  - Cohort 4: HMECs transduced with pRetroX-FLAG-$GNAO1^{R243H}$-IRES-ZsGreen1
- Western blotting will be performed for the following proteins:
  - FLAG
  - $\beta$-ACTIN

## Materials and reagents

| Reagent | Type | Manufacturer | Catalog # | Comments |
|---------|------|--------------|-----------|----------|

*Continued on next page*

*Continued*

| Reagent | Type | Manufacturer | Catalog # | Comments |
|---|---|---|---|---|
| pRetroX-IRES-ZsGreen1 vector | Plasmid | Produced in Protocol 1 | | |
| pRetroX- FLAG-*GNAO1^WT* -IRES-ZsGreen1 vector | Plasmid | Produced in Protocol 1 | | |
| pRetroX-FLAG-*GNAO1^R243H* -IRES-ZsGreen1 vector | Plasmid | Produced in Protocol 1 | | |
| HMECs | Cell line | ATCC | PCS-600-010 | Original product number not specified; Replaces Life Technology brand used in original study |
| HMEC medium | Cell culture | ATCC | PCS-600-03 | Original product number not specified; Replaces Life Technology brand used in original study |
| HMEC supplement | Cell culture | ATCC | PCS-600-040 | Original product number not specified; Replaces Life Technology brand used in original study |
| Bovine pituitary extract | Cell culture | Life Technologies | 13028014 | |
| Penicillin/Streptomycin | Cell culture | Applied Biological Materials | G255 | Original not specified |
| Phoenix amphoteric cells | Cell line | ATCC | ATCC CRL-3213 | Replaces Orbigen brand used in original study |
| DMEM | Cell culture | Sigma | 11965-092 | Original not specified |
| Fetal bovine serum (FBS) | Cell culture | Life Technologies | 12483-020 | Original not specified |
| L-glutamine | Cell culture | Life Technologies | 35050-061 | Original not specified |
| Glucose | Cell culture | Life Technologies | A2494001 | Original not specified |
| Lipofectamine 2000 | Transfection Reagent | Life Technologies | 11668027 | |
| Opti-MEM | Transfection Reagent | Sigma-Aldrich | 31985070 | Original not specified |
| PBS | Buffer | GIBCO | 10010023 | Original not specified |
| 0.45 µm syringe filter | Labware | Millipore | SLHV033RB | Original not specified |
| Trypsin EDTA | Buffer | ABM | TM050 | Original not specified |
| FBS | Buffer | GIBCO | 12483 | Original not specified |
| SDS | Chemical | Left to the discretion of the replicating lab | | |
| 2-mercaptoethanol | Chemical | | | |
| Glycerol | Chemical | | | |
| bromophenol blue | Chemical | | | |
| Tris-HCl | Chemical | | | |
| Bradford Assay | Detection assay | Sigma | B6916-500 ML | Original not specified |
| 12% SDS-PAGE gel | Western Blot Reagent | Invitrogen | EC60252BOX | Original 4–20% |
| OptiProtein Marker | Western Blot Reagent | Applied Biological Materials | G252 | Original not specified |
| PVDF membrane | Western Blot Reagent | Biorad | 162-0015 | Original Nitrocellulose |
| Skim milk powder | Western Blot Reagent | Fisher Scientific | 361021617 | Original not specified |
| 1X TBS solution | Buffer | Fisher Scientific | BP2471-100 | Original not specified |
| Anti-FLAG M2 antibody | Antibody | Sigma | F1804 | |
| Anti-ß-ACTIN antibody | Antibody | Abcam | Ab8227 | Original not specified |
| Anti-mouse HRP-conjugated secondary antibody | Antibody | Abcam | Ab6728 | Original not specified |
| ECL Reagent A and B | Western Blot Reagent | Applied Biological Materials | G075 | Replaces Thermo Fisher brand. |
| X-ray Film | Western Blot Reagent | Kodak | XBT-1 | Original not specified |

## Procedure
### Notes:

- HMECs are grown in complete HMEC medium: HMEC medium supplemented with HMEC supplement, #0.05 mg/mL bovine pituitary extract, 100 U/mL penicillin and 100 mg/ml streptomycin cultured at 37°C and 5% $CO_2$.
- Phoenix cells are grown in complete DMEM medium: DMEM supplemented with 10% (v/v) FBS, 2 mM L-glutamine and 4.5 g/L glucose, 100 U/mL penicillin and 100 mg/ml streptomycin cultured at 37°C and 5% $CO_2$.
- All cells will be sent for mycoplasma testing and STR profiling.

1. #Transfect Phoenix cells with the appropriate retroviral constructs using Lipofectamine 2000 according to manufacturer's instructions.
   a. On the day before transfection, transfer Phoenix cells to fresh medium in 6 well plates and maintain at 37°C and 5% $CO_2$.
   b. On the day of transfection, dilute 2.5 μg plasmid DNA in 500 μl Opti-MEM medium and mix gently.
      i. pRetroX-FLAG-$GNAO1^{WT}$-IRES-ZsGreen1
      ii. pRetroX-FLAG-$GNAO1^{R243H}$-IRES-ZsGreen1
      iii. pRetroX-IRES-ZsGreen1 (empty vector)
   c. Incubate for 30 min at room temperature.
   d. Add DNA-Opti-MEM mixture to 500 μl Lipofectamine 2000.
   e. Add DNA-Lipofectamine LTX complex to wells containing Phoenix cells and mix gently
   f. Incubate cells for 18–48 hr.
      i. Change media after 4–6 hr to complete media containing serum.
   g. Harvest virus-containing supernatants 48 hr post transfection and re-feed cells with DMEM. Incubate at 37°C in a humidified 5% $CO_2$ incubator. Note: Multiple rounds of collection may be required for concentrating stock.
      i. This initial collected media can be stored briefly at 4°C.
   h. After an additional 12–24 hr of culture, collect viral supernatants again and pool with first collection.
   i. #Concentrate viral stock.
      i. Centrifuge the viral supernatant at 3000 rpm for 15 min to remove any cell debris.
      ii. Filter the supernatant through a 0.45 μm syringe filter.
      iii. Ultracentrifuge at 22,000 rpm for 2 hr at 4°C to produce concentrated viral stocks.
      iv. Aliquot virus into screw-cap centrifuge tubes and store at -70°C.
   j. #Titre retrovirus
      i. One day before harvesting viral supernatant, plate $1.2 \times 10^5$ HMECs per well of a 6 well dish.
      ii. On the day of viral supernatant harvesting, count the number of cells in one well to determine cell number at time of infection.
      iii. Add a range of volumes between 2 to 5 μl of concentrated viral supernatant to the wells. Incubate for 72 hr.
      iv. Remove culture medium, wash the wells once with 2 ml PBS.
      v. Add 0.5 ml of 0.25% trypsin EDTA
      vi. Incubate 5 min at 37°C.
      vii. Add 0.5 ml DMEM-10 or 15 (10–15% FBS).
      viii. Pipette up and down with 1 ml pipette and transfer cells to a FACS tube.
      ix. Determine the percentage of GFP-positive cells by FACS analysis.
      x. Calculate the number of transfection units (TU/ml):
         1. Divide the % GFP-positive cells by 100.
         2. Multiply that by the number of cells at the time of infection
         3. Divide that number by the volume of the virus added (ml)
         4. This will yield the number of viral particles per ml.
   k. Use resulting virus to transduce HMECs in Step 3.
2. #One day prior to transduction, seed HMECs in 15 cm plates so they will be 70–90% confluent on the day of transfection.
3. #Transduce HMECs with the appropriate viruses (Optimal MOI will be determined prior to transduction).
   1. Infect HMECs on a 24-well plate with lentivirus.
      i. Cohort 1: Uninfected HMECs [additional negative control]
      ii. Cohort 2: HMECs transduced with pRETRO-IRES-ZsGreen1-empty vector [additional negative control]

 iii. Cohort 3: HMECs transduced with pRETRO-FLAG-*GNAO1^{WT}*-IRES- ZsGreen1
 iv. Cohort 4: HMECs transduced with pRETRO-FLAG-*GNAO1 ^{R243H}*-IRES- ZsGreen1
 2. After 72 hr, check cells under fluorescence microscope to calculate infection rate.
4. #Sterile sort the top 10% of the transduced HMECs by #flow cytometry based on GFP expression.
 a. Trypsinize the cells and resuspend in PBS with 0.5% FBS (FACS buffer)
 b. Pass the cells through the cell strainer to make a single cell suspension.
 c. Sort cells for GFP signal (top 10% selected) on FACS sorter (Influx 100 $\mu$m-18 psi)
 i. 100,000 cells are collected per tube.
5. #Perform Western blots on top 10% GFP positive HMECs to confirm expression of GNAO1:
 a. Spin down the cells for 5 min at maximum speed using an eppendorf tube centrifuge. Aspirate and discard the supernatant.
 b. Add 100 µl to 200 µl of protein lysis buffer depending on the size of the cell pellet.
 i. #Protein lysis buffer: 4% SDS, 10% 2-mercaptoethanol, 20% glycerol, 0.004% bromo-phenol blue, 0.125 M Tris HCl pH 6.8
 ii. Quantify protein concentration using a #Bradford assay according to manufacturer's instructions.
 c. Load equal amounts of total protein in 25 µl sample on a #12% SDS-PAGE gel.
 i. Boil for 7 min before loading
 ii. Load one lane with 8 µl protein marker ladder.
 iii. Run at 150V for 10–15 min.
 iv. When samples reach separation gel, turn to 100V and run for approximately 1.5 hr. Record running time.
 d. Wet transfer to #PVDF membrane #at 95V for 70 min.
 e. #Block membrane with 3% skim milk in Tris-buffered saline (TBS) on shaker for 30 min
 f. Incubate with the following primary antibodies for 1 hr at 37°C:
 i. Mouse Anti-FLAG M2 (1:500 dilution)
 ii. #Mouse Anti-ß-ACTIN (#1:1000 dilution)
 g. Wash membrane 3 times in 1X TBS for 5 min each on shaker.
 h. Incubate with anti-mouse HRP conjugated secondary antibody (#1:1000) for 1 hr on shaker at room temperature.
 i. Remove membrane from secondary antibody and wash three times in 1X TBS for 5 min each.
 j. Prepare ECL solution and incubate membrane.
 k. Expose membrane to X-ray film, develop and scan.

## Deliverables

- Data to be collected:
  - Data for viral titration
  - Flow cytometry data (for viral titration and sorting of transduced cells)
  - Protein determination assay data.
  - Figure 3F: Full scans of all films for each western blot with ladder.
- Sample delivered for further analysis:
  - HMECs transduced with:
    - pRetroX-IRES-ZsGreen1 (empty vector)
    - pRetroX-FLAG-*GNAO1^{WT}*-IRES-ZsGreen1
    - pRetroX-FLAG-*GNAO1^{R243H}*-IRES-ZsGreen1

## Confirmatory analysis plan

- None applicable.

## Known differences from the original study

All known differences are listed in the materials and reagents section above with the originally used item listed in the comments section. All differences have the same capabilities as the original and are not expected to alter the experimental design.

## Provisions for quality control

The cell line used in this experiment will undergo STR profiling to confirm its identity and will be sent for mycoplasma testing to ensure there is no contamination. GNAO1 expression will be confirmed in the top 10% GFP positive HMECs with western blots. All of the raw data, including the analysis files, will be uploaded to the project page on the OSF (https://osf.io/jpeqg/) and made publically available.

## Protocol 3: Anchorage-independent colony formation assay of HMECs transduced with wild-type or mutant GNAO1

This experiment tests the effect of WT or mutant GNAO1 expression on anchorage-independent colony formation of HMECs. It is a replication of the experiments reported in Figure 3D–E.

### Sampling

- Experiment to be repeated a total of 3 times for a power of 99%.
  - See Power Calculations section for details.
- Experiment has 4 cohorts:
  - Cohort 1: Uninfected HMECs [additional negative control]
  - Cohort 2: HMECs transduced with pRetroX-IRES-ZsGreen1-empty vector [additional negative control]
  - Cohort 3: HMECs transduced with pRetroX-FLAG-$GNAO1^{WT}$-IRES-ZsGreen1
  - Cohort 4: HMECs transduced with pRetroX-FLAG-$GNAO1^{R243H}$-IRES-ZsGreen1
- Each cohort will have anchorage independent colony formation quantified.

### Materials and reagents

| Reagent | Type | Manufacturer | Catalog # | Comments |
|---|---|---|---|---|
| HMECs | Cell line | ATCC | PCS-600-010 | Original product number not specified; Replaces Life Technology brand used in original study |
| HMECs transduced with pRetroX-IRES-ZsGreen1-empty vector | Cell line | Produced in Protocol 2 | | |
| HMECs transduced with pRetroX-FLAG-$GNAO1^{WT}$-IRES-ZsGreen1 | Cell line | Produced in Protocol 2 | | |
| HMECs transduced with pRetroX-FLAG-$GNAO1^{R243H}$-IRES-ZsGreen1 | Cell line | Produced in Protocol 2 | | |
| HMEC medium | Cell culture | ATCC | PCS-600-03 | Replaces Life Technology brand used in original study |
| HMEC supplement | Cell culture | ATCC | PCS-600-040 | Original product number not specified; Replaces Life Technology brand used in original study |
| Bovine pituitary extract | Cell culture | Life Technologies | 13028014 | Originl not specified |
| Penicillin/streptomycin | Cell culture | ABM | G255 | Original not specified |
| 6 well plates | Labware | Fisher Scientific | Biolite 12556004 | Original not specified |
| Low melting temperature agar | Cell culture | Bioworld | 40100048-2 | Original not specified |
| Crystal violet | Dye | Left to the discretion of the replicating lab | | Not originally used |
| Methanol (MeOH) | Chemical | | | |
| Acetic acid | Chemical | | | |
| ImageJ | Software | NIH | | Replaces Oxford Optronix GelCount imager and software |

### Procedure

Notes:

- Transduced HMECs are generated in Protocol 2.

1. Grow 3 flasks of transduced and untransduced control HMECs in complete HMEC medium: HMEC medium supplemented with HMEC supplement, #0.5 mg/mL bovine pituitary extract, 100 U/mL penicillin and 100 mg/mL streptomycin cultured at 37°C and 5% $CO_2$ (these will be the biological replicates).
2. Plate a lower layer of #1 ml 0.5% agar per well in twelve wells of 6-well plates.
   a. Let solidify.
3. Suspend 3 wells each of $3 \times 10^4$ HMECs in 1 ml full media containing 0.35% agar containing either:
   a. untransduced HMECs
   b. HMECs transduced with pRetroX-IRES-ZsGreen1 (empty vector)
   c. HMECs transduced with pRetroX-FLAG-*GNAO1*$^{WT}$-IRES-ZsGreen1
   d. HMECs transduced with pRetroX-FLAG-*GNAO1*$^{R243H}$-IRES-ZsGreen1
4. Plate #1 ml suspended cells on top of the lower layer of 0.5% agar in 6-well plates.
5. Incubate the plates for 3 weeks at 37°C and 5% $CO_2$.
   a. #Refresh growth media on top layer every 2–3 days.
6. Assess the presence of colonies.
   a. Stain wells with crystal violet.
      1. Remove media from wells.
      2. Fix with 500 µl of 10% MeOH/10% acetic acid for 10 min.
      3. Remove and stain with 500 µl 0.01% crystal violet for 1 hr.
      4. Remove stain and wash wells.
   b. Image entire well with high-resolution camera.
      1. Include calibration scale in image.
   c. Quantify the number of colonies greater than 200 µm in diameter using ImageJ software.
      1. Set threshold using calibration scale taken during image acquisition.

## Deliverables

- Data to be collected:
  - Figure 3D: Images of colonies.
  - Raw numbers for quantification of colonies for each sample.
  - Figure 3E: Graph of mean number of colonies for each cohort.

## Confirmatory analysis plan

- Statistical Analysis of the Replication Data:
- Note: At the time of analysis we will perform the Shapiro-Wilk test and generate a quantile-quantile plot to assess the normality of the data. We will also perform Levene's test to assess homoscedasticity. If the data appears skewed we will perform the appropriate transformation in order to proceed with the proposed statistical analysis. If this is not possible we will perform the equivalent non-parametric test.
  - Unpaired two-tailed *t*-test of the mean number of colonies in HMECs expressing exogenous GNAO1$^{WT}$ or GNAO1$^{R243H}$.
- Meta-analysis of original and replication attempt effect sizes:
  - This replication attempt will perform the statistical analysis listed above, compute the effects sizes, compare them against the reported effect size in the original paper and use a meta-analytic approach to combine the original and replication effects, which will be presented as a forest plot

## Known differences from the original study

The original study counted cell colonies using GelCount to image, count, and analyze colonies, while the replication attempt will stain with crystal violet to enhance detection of cell colonies, image wells with a high-resolution camera, and use ImageJ software to count and analyze colonies. Since the software and approach used by the original and replication attempt are different, there will likely be some differences in sensitivity and error rates. All known differences are listed in the materials and reagents section above with the originally used item listed in the comments section. All differences have the same capabilities as the original and are not expected to alter the experimental design.

## Provisions for quality control

The cell line used in this experiment will undergo STR profiling to confirm its identity and will be sent for mycoplasma testing to ensure there is no contamination. All of the raw data, including the analysis files, will be uploaded to the project page on the OSF (https://osf.io/jpeqg/) and made publically available.

## Protocol 4: Generation of NIH3T3 cells stably expressing wild-type or mutant MAP2K4

This protocol describes the generation of NIH3T3 cells stably expressing wild-type or mutant MAP2K4 proteins. This protocol also describes verification of expression of MAP2K4 by western blot that will be a replication of Figure 4E. These cells will subsequently be used in Protocols 4 and 5.

### Sampling

- This experiment will be conducted one time to confirm stable expression of exogenous MAP2K4.
- Experiment has 5 cohorts:
  - Cohort 1: Uninfected NIH3T3 cells [additional negative control]
  - Cohort 2: transduced with pRetroX-IRES-ZsGreen1 (empty vector)
  - Cohort 3: transduced with pRetroX-FLAG-$MAP2K4^{WT}$-IRES-ZsGreen1
  - Cohort 4: transduced with pRetroX-FLAG-$MAP2K4^{R228K}$-IRES-ZsGreen1
  - Cohort 5: transduced with pRetroX-FLAG-$MAP2K4^{A279T}$-IRES-ZsGreen1
- To confirm MAP2K4 expression, Western blotting will be performed for the following proteins:
  - FLAG
  - $\beta$-ACTIN [Additional loading control]

### Materials and reagents

| Reagent | Type | Manufacturer | Catalog # | Comments |
|---|---|---|---|---|
| NIH3T3 cells | Cell line | ATCC | CRL-1658 | |
| DMEM medium | Cell culture | Sigma | 11965-092 | Original not specified |
| FBS | Cell culture | Life Technologies | 12483-020 | Original not specified |
| L-glutamine | Cell culture | Life Technologies | 35050-061 | Original not specified |
| Penicillin/Streptomycin | Cell culture | Applied Biological Materials | G255 | Original not specified |
| pRetroX-IRES-ZsGreen1 vector | Plasmid | Produced in Protocol 1 | | |
| pRetroX-FLAG-$MAP2K4^{WT}$-IRES-ZsGreen1 vector | Plasmid | Produced in Protocol 1 | | |
| pRetroX-FLAG-$MAP2K4^{R228K}$-IRES-ZsGreen1 vector | Plasmid | Produced in Protocol 1 | | |
| pRetroX-FLAG-$MAP2K4^{A279T}$-IRES-ZsGreen1 vector | Plasmid | Produced in Protocol 1 | | |
| Phoenix amphoteric cells | Cell line | ATCC | ATCC CRL-3213 | Replaces Orbigen brand used in original study |
| Lipofectamine 2000 | Transfection Reagent | Life Technologies | 11668027 | |
| Opti-MEM | Transfection Reagent | Sigma-Aldrich | 31985070 | Original not specified |
| PBS | Buffer | GIBCO | 10010023 | Original not specified |
| 0.45 µm syringe filter | Labware | Millipore | SLHV033RB | Original not specified |
| Trypsin EDTA | Buffer | ABM | TM050 | Original not specified |
| FBS | Buffer | GIBCO | 12483 | Original not specified |
| SDS | Chemical | | | |
| 2-mercaptoethanol | Chemical | | | |

*Continued on next page*

*Continued*

| Reagent | Type | Manufacturer | Catalog # | Comments |
|---------|------|--------------|-----------|----------|
| Glycerol | Chemical | | | |
| bromophenol blue | Chemical | | | |
| Tris-HCl | Chemical | | | |
| Bradford Assay | Detection assay | Sigma | B6916-500 ML | Original not specified |
| 12% SDS-PAGE gel | Western Blot Reagent | Invitrogen | EC60252BOX | Original 4–20% |
| OptiProtein Marker | Western Blot Reagent | Applied Biological Materials | G252 | Original not specified |
| PVDF membrane | Western Blot Reagent | Biorad | 162-0015 | Original Nitrocellulose |
| 1X TBS solution | Buffer | Fisher Scientific | BP2471-100 | Original not specified |
| Anti-FLAG M2 antibody | Antibody | Sigma | F1804 | |
| Anti-ß-ACTIN antibody | Antibody | Abcam | Ab8227 | Original not specified |
| Anti-mouse HRP-conjugated secondary antibody | Antibody | Abcam | Ab6728 | Original not specified |
| ECL Reagent A and B | Western Blot Reagent | Applied Biological Materials | G075 | Replaces Thermo Fisher brand. |
| X-ray Film | Western Blot Reagent | Kodak | XBT-1 | Original not specified |

## Procedure

Notes:

- NIH3T3 cells are grown in complete DMEM medium: DMEM medium supplemented with 10% (v/v) FBS, 2 mM L-glutamine, 100 U/mL penicillin and 100 mg/mL streptomycin cultured at 37°C and 5% $CO_2$.
- Phoenix cells grown in complete DMEM medium: DMEM supplemented with 10% (v/v) FBS, 2 mM L-glutamine, 100 U/mL penicillin and 100 mg/mL streptomycin cultured at 37°C and 5% $CO_2$.
- All cells will be sent for mycoplasma testing and STR profiling.

1. Transfect Phoenix cells with the appropriate constructs as in Protocol 2 step 1.
2. Transduce NIH3T3 cells with the appropriate viruses as in Protocol 2 steps 2 and
3. Sterile sort the top 10% of the transduced NIH3T3 cells by flow cytometry based on GFP expression as in Protocol 2 Step 4.
4. Perform western blot on sorted cells to confirm expression of MAP2K4 as in Protocol 2 Step 5.

## Deliverables

- Data to be collected:
  - Data for viral titration
  - Flow cytometry data (for viral titration and sorting of transduced cells)
  - Protein determination assay data.
  - Figure 4E: Full scans of all films for each western with ladder.
- Sample delivered for further analysis:
  - NIH3T3 cells transduced with:
    - pRetroX-IRES-ZsGreen1 (empty vector)
    - pRetroX-FLAG-*MAP2K4^{WT}*-IRES-ZsGreen1
    - pRetroX-FLAG-*MAP2K4^{R228K}*-IRES-ZsGreen1
    - pRetroX-FLAG-*MAP2K4^{A279T}*-IRES-ZsGreen1

## Confirmatory analysis plan

- None applicable.

## Known differences from the original study

Not all mutants used in the original study will be replicated. We will not generate MAP2K4 mutations G85R, R154W, N234I or S251N. All known differences are listed in the materials and reagents

section above with the originally used item listed in the comments section. All differences have the same capabilities as the original and are not expected to alter the experimental design.

## Provisions for quality control

The cell lines used in this experiment will undergo STR profiling to confirm their identity and will be sent for mycoplasma testing to ensure there is no contamination. MAP2K4 expression will be confirmed in the top 10% GFP positive HMECs with Western blots. All of the raw data, including the analysis files, will be uploaded to the project page on the OSF (https://osf.io/jpeqg/) and made publically available.

## Protocol 5: Anchorage-independent colony formation assay of NIH3T3 cells transduced with wild-type or mutant MAP2K4

This experiment tests the effect of WT or mutant MAP2K4 expression on anchorage-independent colony formation of NIH3T3 cells. It is a replication of the experiments reported in Figure 4C and 4D.

### Sampling

- Experiment to be repeated a total of 3 times for a minimum power of 99%.
  - See Power Calculations section for details.
- Experiment has 5 (generated in Protocol 4) cohorts:
  - Cohort 1: Uninfected NIH3T3 cells [additional negative control]
  - Cohort 2: NIH3T3 cells transduced with with pRetroX-IRES-ZsGreen1-empty vector
  - Cohort 3: NIH3T3 cells transduced with with pRetroX-FLAG-$MAP2K4^{WT}$-IRES-ZsGreen1
  - Cohort 4: NIH3T3 cells transduced with with pRetroX-FLAG-$MAP2K4^{R228K}$-IRES-ZsGreen1
  - Cohort 5: NIH3T3 cells transduced with with pRetroX-FLAG-$MAP2K4^{A279T}$-IRES-ZsGreen1
- Each cohort will have anchorage independent colony formation quantified.

### Materials and reagents

| Reagent | Type | Manufacturer | Catalog # | Comments |
|---------|------|--------------|-----------|----------|
| NIH3T3 cells transduced with pRetroX-IRES-ZsGreen1-empty vector | Cell line | Produced in Protocol 4 | | |
| NIH3T3 cells transduced with pRetroX-FLAG-$MAP2K4^{WT}$-IRES-ZsGreen1 | Cell line | Produced in Protocol 4 | | |
| NIH3T3 cells transduced with pRetroX-FLAG-$MAP2K4^{R228K}$-IRES-ZsGreen1 | Cell line | Produced in Protocol 4 | | |
| NIH3T3 cells transduced with pRetroX-FLAG-$MAP2K4^{A279T}$-IRES-ZsGreen1 | Cell line | Produced in Protocol 4 | | |
| DMEM medium | Cell culture | Sigma | 11965-092 | Original not specified |
| FBS | Cell culture | Life Technologies | 12483-020 | Original not specified |
| L-glutamine | Cell culture | Life Technologies | 35050-061 | Original not specified |
| Penicillin/streptomycin | Cell culture | Applied Biological Materials | G255 | Original not specified |
| 6 well plates | Labware | Fisher Scientific | Biolite 12556004 | Original not specified |
| Low melting temperature agar | Cell culture | Bioworld | 40100048-2 | Original not specified |
| Crystal violet | Dye | Left to the discretion of the replicating lab | | Not originally used |
| Methanol (MeOH) | Chemical | | | |
| Acetic acid | Chemical | | | |
| ImageJ | Software | NIH | | Replaces Oxford Optronix GelCount imager and software |

### Procedure
Note:

- All cells will be sent for mycoplasma testing and STR profiling.

1. Grow 3 flasks each of transduced NIH3T3 cells generated in Protocol 4 in complete DMEM medium: DMEM medium supplemented with 10% (v/v) FBS, 2 mM L-glutamine, 100 U/mL penicillin and 100 mg/mL streptomycin cultured at 37°C and 5% $CO_2$. (these will be the biological replicates)
2. Plate a lower layer of [#]1 ml 0.5% agar per well in 16 wells of 6-well plates.
   a. Let solidify.
3. Suspend 3 plates each of $1 \times 10^4$ cells/plate in 1 ml full media containing 0.35% agar containing either:
   - uninfected NIH3T3 cells [additional negative control]
   - NIH3T3 cells transduced with pRetroX-IRES-ZsGreen1-empty vector
   - NIH3T3 cells transduced with pRetroX-FLAG-$MAP2K4^{WT}$-IRES-ZsGreen1
   - NIH3T3 cells transduced with pRetroX-FLAG-$MAP2K4^{R228K}$-IRES-ZsGreen1
   - NIH3T3 cells transduced with pRetroX-FLAG-$MAP2K4^{A279T}$-IRES-ZsGreen1
4. Plate [#]1 ml suspended cells on top of the lower layer of 0.5% agar in 6-well plates.
5. Incubate the plates for 3 weeks at 37°C and 5% $CO_2$.
   a. [#]Refresh growth media from top layer every 2–3 days.
6. Assess the presence of colonies.
   a. Stain wells with crystal violet.
      1. Remove media from wells.
      2. Fix with 500 µl of 10% MeOH/10% acetic acid for 10 min.
      3. Remove and stain with 500 µl 0.01% crystal violet for 1 hr.
      4. Remove stain and wash wells.
   b. Image entire well with a high-resolution camera.
      1. Include calibration scale in image.
   c. Quantify the number of colonies greater than 100 µm in diameter using ImageJ software.
      1. Set threshold using scale taken during image acquisition.

## Deliverables

- Data to be collected:
  - Figure 4C: Images of colonies.
  - Raw numbers for quantification of colonies for each sample.
  - Figure 4D: Graph of mean number of colonies for each cohort.

## Confirmatory analysis plan

- Statistical Analysis of the Replication Data:
- Note: At the time of analysis we will perform the Shapiro-Wilk test and generate a quantile-quantile plot to assess the normality of the data. We will also perform Levene's test to assess homoscedasticity. If the data appears skewed we will perform the appropriate transformation in order to proceed with the proposed statistical analysis. If this is not possible we will perform the equivalent non-parametric test.
  - One-way ANOVA of the mean number of colonies in NIH3T3 cells expressing exogenous $MAP2K4^{WT}$, $MAP2K4^{R228K}$, or $MAP2K4^{A279T}$ followed by planned comparisons using Fisher's LSD:
    - $MAP2K4^{WT}$ vs $MAP2K4^{R228K}$
    - $MAP2K4^{WT}$ vs $MAP2K4^{A279T}$
- Meta-analysis of original and replication attempt effect sizes:
  - Compute the effect sizes of each comparison, compare them against the effect size in the original paper and use a random effects meta-analytic approach to combine the original and replication effects, which will be presented as a forest plot.

## Known differences from the original study

Not all mutants used in the original study will be replicated. We will not generate MAP2K4 mutations G85R, R154W, N234I or S251N. The original study counted cell colonies using GelCount to image, count, and analyze colonies, while the replication attempt will stain with crystal violet to enhance detection of cell colonies, image wells with a high-resolution camera, and use ImageJ software to

count and analyze colonies. Since the software and approach used by the original and replication attempt are different, there will likely be some differences in sensitivity and error rates. All known differences are listed in the materials and reagents section above with the originally used item listed in the comments section. All differences have the same capabilities as the original and are not expected to alter the experimental design.

## Provisions for quality control

The cell line used in this experiment will undergo STR profiling to confirm its identity and will be sent for mycoplasma testing to ensure there is no contamination. All of the raw data, including the analysis files, will be uploaded to the project page on the OSF (https://osf.io/jpeqg/) and made publically available.

## Protocol 6: Assessing the kinase activity of wild-type or mutant MAP2K4

This experiment tests the *in vitro* kinase activity of WT or mutant MAP2K4 immunoprecipitated from NIH3T3 cells. It is a replication of the experiment reported in Figure 4F.

### Sampling

- Experiment to be repeated a total of 4 times.
  - The original data is qualitative, thus to determine an appropriate number of replicates to initially perform, sample sizes were determined based on a range of potential variance.
    - See Power Calculations section for details.
- Experiment has 5 cohorts:
  - Cohort 1: Uninfected NIH3T3 cells [additional negative control]
  - Cohort 2: NIH3T3 cells transduced with pRetroX-IRES-ZsGreen1 (empty vector)
  - Cohort 3: NIH3T3 cells transduced with pRetroX-FLAG-*MAP2K4^{WT}*-IRES-ZsGreen1
  - Cohort 4: NIH3T3 cells transduced pRetroX-FLAG-*MAP2K4^{R228K}*-IRES-ZsGreen1
  - Cohort 5: NIH3T3 cells transduced with pRetroX-FLAG-*MAP2K4^{A279T}*-IRES-ZsGreen1
- A kinase assay is performed for each cohort using the following substrates:
  - Myelin basic protein (MBP)
  - Inactive MAPK9/JNK2

### Materials and reagents

| Reagent | Type | Manufacturer | Catalog # | Comments |
|---------|------|--------------|-----------|----------|
| NIH3T3 cells transduced with pRetroX-IRES-ZsGreen1-empty vector | Cell line | Produced in Protocol 4 | | |
| NIH3T3 cells transduced with pRetroX-FLAG-*MAP2K4^{WT}*-IRES-ZsGreen1 | Cell line | Produced in Protocol 4 | | |
| NIH3T3 cells transduced with pRetroX-FLAG-*MAP2K4^{R228K}*-IRES-ZsGreen1 | Cell line | Produced in Protocol 4 | | |
| NIH3T3 cells transduced with pRetroX-FLAG-*MAP2K4^{A279T}*-IRES-ZsGreen1 | Cell line | Produced in Protocol 4 | | |
| DMEM medium | Cell culture | Sigma | 11965-092 | Original not specified |
| FBS | Cell culture | Life Technologies | 12483-020 | Original not specified |
| L-glutamine | Cell culture | Life Technologies | 35050-061 | Original not specified |
| EZview FLAG-M2-antibody-coupled affinity gel | Chromatography | Sigma | A2220 | |
| Penicillin/streptomycin | Cell culture | Applied Biological Materials | G255 | Original not specified |
| Cell Lysis Buffer | Buffer | Cell Signaling Technology | 9803 | Original product number not specified |

*Continued on next page*

*Continued*

| Reagent | Type | Manufacturer | Catalog # | Comments |
|---|---|---|---|---|
| Phenylmethanesulfonyl Fluoride (PMSF) | Protease inhibitor | Cell Signaling Technology | 8853 | Original not specified |
| Myelin basic protein | Protein | Signalchem | M42-51N | Replaces Millipore AB15542 |
| Inactive MAPK9/JNK2 | Protein | Invitrogen | PV3621 | Listed as MAP2K7 in original paper. |
| [γ-32P]ATP | Chemical | Perkin Elmer | BLU002H250UC | |
| Kinase Reaction Buffer | Buffer | Cell Signaling Technology | 9802 | Original product number not specified |
| Anti-FLAG M2 Magnetic Beads | Kinase assay reagent | Sigma-Aldrich | M8823 | Original not specified |
| 12% SDS-PAGE | Western Blot Reagent | Invitrogen | EC60252BOX | Original 4–20% |
| Bradford Assay | Detection assay | Sigma | B6916-500 ML | Original not specified |
| OptiProtein Marker | Western Blot Reagent | Applied Biological Materials | G252 | Original not specified |
| PVDF membrane | Western Blot Reagent | Biorad | 162-0015 | Original Nitrocellulose |
| Skim milk powder | Western Blot Reagent | Fisher Scientific | 361021617 | Original not specified |
| 1X TBS solution | Buffer | Fisher Scientific | BP2471-100 | Original not specified |
| Anti-FLAG M2 antibody | Antibody | Sigma | F1804 | |
| Anti-ß-ACTIN antibody | Antibody | Abcam | Ab8227 | Original not specified |
| Anti-mouse HRP-conjugated secondary antibody | Antibody | Abcam | Ab6728 | Original not specified |
| ECL Reagent A and B | Western Blot Reagent | Applied Biological Materials | G075 | Replaces Thermo Fisher brand. |
| X-ray Film | Western Blot Reagent | Kodak | XBT-1 | Original not specified |

## Procedure

Note:

- Transduced NIH3T3 cells are generated in Protocol 4.
- All cells will be sent for mycoplasma testing and STR profiling.

1. Grow 4 flasks of NIH3T3 in complete DMEM medium: DMEM medium supplemented with 10% (v/v) FBS, 2 mM L-glutamine, 100 U/mL penicillin and 100 μg/mL streptomycin cultured at 37°C and 5% $CO_2$. These are the biological replicates.
2. Generate cell lysates:
   a. Plate cells for kinase assay so they will be 70–90% confluent on the day of harvest.
   b. Replace with serum free media (0% FBS) and serum starve cells for 24 hr.
   c. Wash cells with PBS and lyse with Cell Lysis Buffer (with 1 mM PMSF added just before use).
   d. Clarify lysates.
   e. Quantify protein concentration using a Bradford Assay according to manufacturer's instructions.
   f. Adjust samples to equalize for total amount of protein and concentration.
3. Perform immunoprecipitation:
   a. Incubate clarified lysates with anti-FLAG-M2 conjugated beads overnight at 4°C.
   b. Spin beads down at 10,000x*g* for 30 s, remove supernatant and wash three times with Cell lysis buffer (with 1 mM PMSF added just before use).
   c. Remove sample for input analysis and divide sample equally between two microcentrifuge tubes.
   d. Spin beads down and remove supernatant.
4. Kinase Assay
   a. Add 25 μl of Kinase Reaction Buffer supplemented with 10 μM ATP and 2 μCi [γ-32P]ATP with either:
      i. Myelin basic protein (MBP) ([#]1:2000)
      ii. Inactive MAPK9/JNK2 ([#]1:2000).

 b. Incubate samples for 30 min at 30°C.

 c. Stop kinase reactions by adding SDS sample buffer.

 d. Resolve kinase reactions on [#]15% SDS-PAGE gel with protein ladder.

 e. [#]Fix gel for 15 min in 5% methanol, 7% acetic acid and dry at 60°C for 30 min.

 f. Expose gel to X-ray film and scan images.

5. Perform western blot on input sample for MAP2K4 expression as in Protocol 2 Step 5.

6. For each replicate normalize each protein (MBP and MAPK9/JNK2) to MAP2K4 input levels and then normalize each sample to MAP2K4$^{WT}$.

## Deliverables

- Data to be collected:
  - Protein determination assay data.
  - Figure 4F: Full images of autoradiographs for each kinase assay substrate with ladder.
  - Figure 4F: Scans of full films for western blot of MAP2K4 input with ladder.

## Confirmatory analysis plan

- Statistical Analysis of the Replication Data:
- Note: At the time of analysis we will perform the Shapiro-Wilk test and generate a quantile-quantile plot to assess the normality of the data. We will also perform Levene's test to assess homoscedasticity. If the data appears skewed we will perform the appropriate transformation in order to proceed with the proposed statistical analysis. If this is not possible we will perform the equivalent non-parametric test.
  - Bonferroni corrected one-sample $t$-tests of normalized pMBP levels from the following MAP2K4 variants compared to 1 (MAP2K4$^{WT}$):
    - MAP2K4$^{R228K}$
    - MAP2K4$^{A279T}$
  - Bonferroni corrected one-sample $t$-tests of normalized pMAPK9/pJNK levels from the following MAP2K4 variants compared to 1 (MAP2K4$^{WT}$):
    - MAP2K4$^{R228K}$
    - MAP2K4$^{A279T}$
- Meta-analysis of effect sizes:
  - Since some of the band intensities in the original paper were unable to be quantified the replication study will record and make accessible all autoradiographs collected. This will allow for a subjective comparison of the original images and the replication images. Additionally, the replication will quantify the results in an additional exploratory measure. This cannot be compared to the original reported results, but will be presented to understand the utility of analyzing the data in a quantitative manner.

## Known differences from the original study

Not all mutants used in the original study will be replicated. We will not generate MAP2K4 mutations G85R, R154W, N234I or S251N. All known differences are listed in the materials and reagents section above with the originally used item listed in the comments section. All differences have the same capabilities as the original and are not expected to alter the experimental design.

## Provisions for quality control

The cell line used in this experiment will undergo STR profiling to confirm its identity and will be sent for mycoplasma testing to ensure there is no contamination. All of the raw data, including the analysis files, will be uploaded to the project page on the OSF (https://osf.io/jpeqg/) and made publicly available.

## Power calculations

For additional details on power calculations, please see analysis scripts and associated files on the Open Science Framework:

https://osf.io/bxr2d/

## Protocol 1:

- Not applicable

## Protocol 2:

- Not applicable

## Protocol 3:

### Summary of original data

- Note; values are from data shared by authors, which was reported in Figure 3E.

| Vector | Mean # of colonies >200 μm diameter | Stdev | N |
|---|---|---|---|
| WT | 113.5 | 10.607 | 2 |
| R243H | 201.5 | 16.263 | 2 |

### Test family

- Two-tailed *t* test, difference between two independent means, alpha error = 0.050

### Power calculations

- Performed with G*Power software, version 3.1.7 (*Faul et al., 2007*).

| Group 1 | Group 2 | Effect size *d* | A priori power | Group 1 sample size | Group 2 sample size |
|---|---|---|---|---|---|
| WT | R243H | 6.40954 | 87.2%[1,2] | 2[1] | 2[1] |

[1] 3 samples per group will be used making the achieved power of 99.9%.
[2] The calculation was also performed with the non-parametric Wilcoxon-Mann-Whitney test, which gives an achieved power of 99.9% with a sample size of 3 per group.

## Protocol 4:

- Not applicable

## Protocol 5:

### Summary of original data

- Note: values are from data shared by authors, which was reported in Figure 4D:

| Vector | Mean # of colonies >100 μm diameter | Stdev | N |
|---|---|---|---|
| WT | 16 | 2.8284 | 2 |
| R228K | 36 | 2.8284 | 2 |
| A279T | 94.5 | 7.7782 | 2 |

## Test family

- ANOVA: Fixed effects, omnibus, one-way, alpha error = 0.05

## Power calculations

- Performed with G*Power software, version 3.1.7 (*Faul et al., 2007*).
- ANOVA F test statistic and partial $\eta^2$ performed with R software, version 3.1.2 (*Team, 2014*).

| Groups | F test statistic | Partial $\eta^2$ | Effect size f | A priori power | Total sample size |
|---|---|---|---|---|---|
| NIH3T3 cells transduced with WT or *MAP2K4* mutants | $F_{(2,3)}$ = 130.52 | 0.98864 | 9.3280 | 99.9% | 6[1] (3 groups) |

[1] 9 total samples (3 per group) will be used as a minimum sample size making the power 99.9%.

## Test family

- Two-tailed *t* test, difference between two independent means, Fisher's LSD: alpha error = 0.05

## Power calculations

- Performed with G*Power software, version 3.1.7 (*Faul et al., 2007*).

| Group 1 | Group 2 | Effect size d | A priori power | Group 1 sample size | Group 2 sample size |
|---|---|---|---|---|---|
| WT | R228K | 7.07107 | 99.9%[1,2] | 2[1] | 2[1] |
| WT | A279T | 13.4134 | 99.9%[1,2] | 2[1] | 2[1] |

[1] 3 samples per group will be used making the power 99.9%.
[2] The calculation was also performed with the non-parametric Wilcoxon-Mann-Whitney test, which gives an achieved power of 99.9% with a sample size of 3 per group.

## Protocol 6
### Summary of original data

- Note: data estimated from the image reported in Figure 4F.
  - The original data presented is qualitative (images of Western blots). We used ImageJ version 1.50a (*Schneider et al., 2012*) to perform densitometric analysis of the presented bands to quantify the original effect size where possible. The data presented in Figure 4F for Input MAP2K4 were unable to be quantified for all bands and were thus excluded from the normalization. Additionally, the WT values provide under-estimates of the actual values since the WT bands were saturated and unable to be quantified.

| Variant | Normalized pJNK band intensity to WT | Normalized pMBP band intensity to WT |
|---|---|---|
| WT | 1 | 1 |
| R228K | 0.299736 | 0.057556 |
| A279T | 0.613378 | 0.096804 |

- The original data does not indicate the error associated with multiple biological replicates. To identify a suitable sample size, power calculations were performed using different levels of relative variance.

## Test family

- *t*-test: Means: Difference from constant (one sample case): Bonferroni's correction: alpha error = 0.0125.

## Power calculations

- Performed with G*Power software, version 3.1.7 (*Faul et al., 2007*).
- 2% variance

| Substrate | Variant | Constant (WT) | Effect size *d* | A priori power | Sample size per group |
|-----------|---------|---------------|-----------------|----------------|------------------------|
| P-JNK | R228K | 1 | 116.813 | 99.9% | 3 |
| | A279T | 1 | 31.5157 | 99.9% | 3 |
| P-MBP | R228K | 1 | 818.713 | 99.9% | 3 |
| | A279T | 1 | 466.507 | 99.9% | 3 |

- 15% variance

| Substrate | Variant | Constant (WT) | Effect size *d* | A priori power | Sample size per group |
|-----------|---------|---------------|-----------------|----------------|------------------------|
| P-JNK | R228K | 1 | 15.5751 | 99.9% | 3 |
| | A279T | 1 | 4.20210 | 92.2% | 4 |
| P-MBP | R228K | 1 | 109.162 | 99.9% | 3 |
| | A279T | 1 | 62.2009 | 99.9% | 3 |

- 28% variance

| Substrate | Variant | Constant (WT) | Effect size *d* | A priori power | Sample size per group |
|-----------|---------|---------------|-----------------|----------------|------------------------|
| P-JNK | R228K | 1 | 8.34380 | 92.6% | 3 |
| | A279T | 1 | 2.25112 | 88.7% | 6 |
| P-MBP | R228K | 1 | 58.4795 | 99.9% | 3 |
| | A279T | 1 | 33.3219 | 99.9% | 3 |

- 40% variance

| Substrate | Variant | Constant (WT) | Effect size *d* | A priori power | Sample size per group |
|-----------|---------|---------------|-----------------|----------------|------------------------|
| P-JNK | R228K | 1 | 5.84066 | 99.5% | 4 |
| | A279T | 1 | 1.57579 | 82.5% | 8 |
| P-MBP | R228K | 1 | 40.9357 | 99.9% | 3 |
| | A279T | 1 | 23.3254 | 99.9% | 3 |

- Based on these ranges of variance, which use a conservative effect size estimate since the original data were unable to be quantified, we will run the experiment four times.

## Acknowledgements

The Reproducibility Project: Cancer Biology core team would like to thank the original authors, in particular Bijay Jaiswal, for generously sharing critical information to ensure the fidelity and quality of this replication attempt. We are grateful to Courtney Soderberg at the Center for Open Science for assistance with statistical analyses. We would also like to thank the following companies for generously donating reagents to the Reproducibility Project: Cancer Biology; American Type Culture Collection (ATCC), Applied Biological Materials, BioLegend, Charles River Laboratories, Corning Incorporated, DDC Medical, EMD Millipore, Harlan Laboratories, LI-COR Biosciences, Mirus Bio, Novus Biologicals, Sigma-Aldrich, and System Biosciences (SBI).

## Additional information

### Group author details

Reproducibility Project: Cancer Biology

Elizabeth Iorns: Science Exchange, Palo Alto, United States; William Gunn: Mendeley, London, United Kingdom; Fraser Tan: Science Exchange, Palo Alto, United States; Joelle Lomax: Science Exchange, Palo Alto, United States; Nicole Perfito: Science Exchange, Palo Alot, United States; Timothy Errington: Center for Open Science, Charlottesville, United States

### Competing interests

VS, LY the experiments presented in this manuscript will be conducted at Applied Biological Materials, which is a Science Exchange lab. RP:CB employed by and holds shares in Science Exchange Inc. The other authors declare that no competing interests exist.

### Funding

| Funder | Author |
| --- | --- |
| Laura and John Arnold Foundation | The Reproducibility Project: Cancer Biology Core Team |

The Reproducibility Project: Cancer Biology is funded by the Laura and John Arnold Foundation, provided to the Center for Open Science in collaboration with Science Exchange. The funders had no role in study design, data collection and interpretation, or the decision to submit the work for publication.

### Author contributions

VS, LY, ABA, KO, Drafting or revising the article; RP:CB, Conception and design, Drafting or revising the article

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
