## [Decision Letter]

Thank you for submitting your work entitled "Registered report: Diverse somatic mutation patterns and pathway alterations in human cancers" for consideration by *eLife*. Your submission has been evaluated by a Senior Editor, who would like you to revise the article as follows prior to peer review:

1) In the case of MAP2K4 there has been independent validation by others that the mutants lack kinase activity, and can drive transformation. In this regard, the authors have missed an important paper (Ahn et al. MCB 31:4270), where the authors examined the properties of 11 MAP2K4 cancer mutants for kinase activity and their ability to affect cell migration and invasion in culture, and alter autochthonous mutant KRas driven NSCLC tumor growth in vivo, reaching the conclusion that MAP2K4 acts as a tumor suppressor. We suggest that the authors add discussion of the Ahn et al. paper and review the literature on MAP2K4 mutations both prior to and subsequent to the Kan et al. paper in question, comprehensively, to ensure that there aren't any other omissions.

2) A technical issue relates to the authors' proposed use of inactive MAP2K7/JNK2 as a substrate for the in vitro MAP2K4 kinase assay. MAP2K7 (aka MKK7) is not the same as JNK2, and it is not a substrate for MAP2K4. This notation came from the original Kan et al. paper, but I suspect they mean MAPK9/JNK2, which is what the Invitrogen website lists under the PV3621 catalogue number they give. Could you check and correct this accordingly?

[Editors' note: further revisions were requested prior to acceptance, as described below.]

Thank you for submitting your work entitled "Registered report: Diverse somatic mutation patterns and pathway alterations in human cancers" for consideration by *eLife*. Your article has been reviewed by three peer reviewers, and the evaluation has been overseen by Tony Hunter as the Senior Editor and Reviewing Editor. Two of the three reviewers, Somasekar Seshagiri and John Brognard, have agreed to share their names.

The reviewers have discussed the reviews with one another and the Reviewing editor has drafted this decision to help you prepare a revised submission.

Summary:

As part of the Reproducibility Project: Cancer Biology, the authors will set out to replicate experimental data on mutant forms of GNAO1 and MAP2K4 that were published in Kan et al. (Nature 2010). The authors note that the functional relevance of the mutant forms of GNAO1 and MAP2K4 in cancer has been established by subsequent studies and propose to replicate the colony formation assay and kinase assay (MAP2K4) experiments reported in Kan et al. (Nature 2010). The authors describe the reagent generation, experimental work and analysis in great detail. The reviewers agree that the proposed plan is appropriate and closely adheres to published methods, but we have the following requests for revisions before publication of the Registered Report.

Essential revisions:

In the Registered Report, the authors use six protocols to replicate the experiments. For protocol 3, 5 and 6 power calculations and confirmatory analysis plans are described. In the confirmatory analysis plan of the protocol five One-way Anova is followed by planned comparisons using Fisher's LSD and in the corresponding power calculations for the t-tests a 0.05 α error is used. Fisher's LSD, however, does not control the family wise error rate (see Hayter, 1986) and it is useful only for the calculation of the effect size d. A Bonferroni correction should be used (α = 0.025. This issue needs to be addressed.

Reference: Anthony J Hayter. The maximum familywise error rate of Fisher's least significant difference test. Journal of the American Statistical Association, 81(396): 1000-1004, 1986. doi: 10.1080/01621459.1986.10478364

---

## [Author Response]

*1) In the case of MAP2K4 there has been independent validation by others that the mutants lack kinase activity, and can drive transformation. In this regard, the authors have missed an important paper (Ahn et al. MCB 31:4270), where the authors examined the properties of 11 MAP2K4 cancer mutants for kinase activity and their ability to affect cell migration and invasion in culture, and alter autochthonous mutant KRas driven NSCLC tumor growth* in vivo*, reaching the conclusion that MAP2K4 acts as a tumor suppressor. We suggest that the authors add discussion of the Ahn* et al.

*paper and review the literature on MAP2K4 mutations both prior to and subsequent to the Kan et al. paper in question, comprehensively, to ensure that there aren't any other omissions.*

We have incorporated a discussion of this study (Ahn et al. 2011) into the Introduction. We have also added other citations to expand the state of knowledge about the function of MAP2K4 mutations.

*2) A technical issue relates to the authors' proposed use of inactive MAP2K7/JNK2 as a substrate for the* in vitro *MAP2K4 kinase assay. MAP2K7 (aka MKK7) is not the same as JNK2, and it is not a substrate for MAP2K4. This notation came from the original Kan et al. paper, but I suspect they mean MAPK9/JNK2, which is what the Invitrogen website lists under the PV3621 catalogue number they give. Could you check and correct this accordingly?*

We have made this change. The Invitrogen PV3621 MAPK9 is listed correctly listed in the reagents list.

[Editors' note: further revisions were requested prior to acceptance, as described below.]

Essential revisions: In the Registered Report, the authors use six protocols to replicate the experiments. For protocol 3, 5 and 6 power calculations and confirmatory analysis plans are described. In the confirmatory analysis plan of the protocol five One-way Anova is followed by planned comparisons using Fisher's LSD and in the corresponding power calculations for the t-tests a 0.05 α error is used. Fisher's LSD, however, does not control the family wise error rate (see Hayter, 1986) and it is useful only for the calculation of the effect size d. A Bonferroni correction should be used (α

*= 0.025. This issue needs to be addressed. Reference: Anthony J Hayter. The maximum familywise error rate of Fisher's least significant difference test. Journal of the American Statistical Association, 81(396): 1000-1004, 1986. doi: 10.1080/01621459.1986.10478364*

We agree with the reviewers’ comment on the use of a correction, such as Bonferroni or the modification of LSD by Hayter as ways to control for the MFWER; however as Hayter describes in his 1986 paper, this applies in situations where the ANOVA is unbalanced or with a balanced design with four or more populations. Since the proposed analysis is balanced with three population groups, the LSD is sufficiently conservative and powerful to account for the multiple comparisons in this specific situation. This is further explained by Levin et al., 1994 and discussed in Maxwell and Delaney, 2004 (Chapter 5) and Cohen, 2001 (Chapter 12).

References:

Levin, J.R., Serline, R.C., & Seaman M.A. (1994). A controlled, powerful multiple-comparison strategy for several situations. Psychological Bulletin*, 115*, 153-159.

Maxwell, S.E. & Delaney, H.D. (2004). Designing experiments and analyzing data: a model comparison perspecitive. Lawrence Erlbaum Associates, Mahwah, N.J., 2^nd^ edition.

Cohen, B.H. (2001). Explaining psychological statistics. John Wiley and Sons, New York, 2^nd^ edition.